# Applicability Evaluation of Surface and Sub-Surface Defects for Railway Wheel Material Using Induced Alternating Current Potential Drops

**DOI:** 10.3390/s22249981

**Published:** 2022-12-18

**Authors:** Seok-Jin Kwon, Jung-Won Seo, Min-Soo Kim, Young-Sam Ham

**Affiliations:** Advanced Railroad Vehicle Division, Korea Railroad Research Institute, #176, Cheoldo Bangmulgwan-ro, Uiwang-si 16105, Gyeonggi-do, Republic of Korea

**Keywords:** railway wheel, non-destructive evaluation, induced AC potential drop, detection sensor

## Abstract

The majority of catastrophic wheelset failures are caused by surface opening fatigue cracks in either the wheel tread or wheel inner. Since failures in railway wheelsets can cause disasters, regular inspections to check for defects in wheels and axles are mandatory. Currently, ultrasonic testing, acoustic emissions, and the eddy current testing method are regularly used to check railway wheelsets in service. Yet, in many cases, despite surface and subsurface defects of the railroad wheels developing, the defects are not clearly detected by the conventional non-destructive inspection system. In the present study, a new technique was applied to the detection of surface and subsurface defects in railway wheel material. The results indicate that the technique can detect surface and subsurface defects of railway wheel specimens using the distribution of the alternating current (AC) electromagnetic field. In the wheelset cases presented, surface cracks with depths of 0.5 mm could be detected using this method.

## 1. Introduction

During the operation of railway vehicles, damaged wheels can lead to the deterioration of ride comfort during running and an increase in maintenance costs. It is very important to evaluate wheel defects with high sensitivity. The most frequent defects in wheels occur because of rolling contact fatigue caused by contact with rails and thermal stress caused by brake fiction heat [1,2,3,4]. In addition, it is known that defects inside wheels have a huge effect on the surface damage of wheels [5]. Rainer et al. reported a technique that detects defects in and outside of wheels by applying a new ultrasonic technology and an eddy current technology, which they tested on Deutsche Bahn [6]. This method has the advantage of detecting radial and tangential defects inside wheels and also detecting defects on the surface of wheels. Richard et al., meanwhile, studied the detection of wheel defects using the laser–air hybrid ultrasonic technique [7]. This method has the advantage of detecting surface defects of wheels when a railway vehicle comes into the depot and has a defect detection probability of 90% or more. Hwang et al. conducted research on the applicability of a differential type of hall sensor to detect defects in wheels for a high-speed railway and showed that it is possible to detect surface defects in wheel specimens using a new non-destructive test method [8]. Lee et al. performed research to detect surface defects in wheel treads using an electromagnetic acoustic transducer [9]. By designing the EMAT angle so that it can be adjusted in order to ensure that it comes into contact with wheel treads, the researchers accurately detected defects on the surface. For the detection of the defects inside wheels for railway vehicles, the ultrasonic test method is used, and for more accurate defect detection, the array method is applied [10,11,12]. In addition, to detect surface defects, a hybrid method combining the eddy current and laser methods is used [13,14,15].

Recently, studies on the applicability of the relevant technologies in different fields have been performed in line with the advancement of various non-destructive test technologies [16,17,18,19,20]. Alemi et al. reported extensively on condition monitoring techniques for railway wheels in on-board and wayside measurement [21]. The authors stated that research on the detection accuracy and monitoring for sub-surface defects is necessary. Shen et al. discussed the evaluation of surface fatigue cracks using alternating current field measurement (ACFM) in the rail sector [22].

There are many defect types on the surface of railway wheels; large defects are removed at the depot through wheel reprofiling during regular inspection, but the detection possibility for sub-surface defects in wheels is much lower.

The purpose of the present paper is to evaluate the possibility of detecting surface and subsurface defects of railway wheel material using the distribution of the induced alternating current (AC) electromagnetic field.

## 2. Application of Induced AC Potential Drop

### 2.1. Principle of Crack Detection

In the early 1990s, Dorver et al. developed the AC potential drop technique, which can hyper-sensitively detect surface defects, predict their shape, and continuously detect crack propagation. This method uses the skin effect, where the electric current flows mainly near the surface when supplying a constant current AC power supply sized to match the measurement object. When a surface defect exists, the electric current flows along the defect surface, and because the flow course of the electric current changes according to the depth of the defect, the surface defect can be detected. However, the electric current distribution on the surface created by the skin effect changes according to the distance from a supply unit, and this causes problems, such as the ground connection problem of the contact point and an error in the measured value depending on the degree of contamination of the measurement surface. To deal with the problem of the electric current distribution between supply units, among other problems, the induced AC potential drop (ACPD) technique was developed to concentrate the measuring current on the detection area. In the conventional ACPD method, the current flows through the specimen. In the induced alternating current potential drop (IACPD) method, the current flows through an induction wire and the induced current is detected. Table 1 presents a comparison of potential drop techniques.

The induced AC potential drop technique is a new non-destructive testing technique that can detect defects in railway wheels by applying an electromagnetic field and examining the potential drop variation. The principle of the induced AC potential drop technique is shown in Figure 1, where AC (I) is supplied to a conducting wire, creating a magnetic field (H) around the wire at a right angle to the electric current direction. Since this magnetic field changes with time, just like the frequency of the electric current, it induces an electric current through the measurement object and generates an electromotive force (EMF). This electric current intensively flows with a greater value directly below the wire compared to other areas. Although the induced electric current flows in the opposite direction to the supply power, it has the same skin effect as the supply power. Because the focused electric current flows along the defect surface, using a measurement system to measure how the electric current changes according to defects enables information on the defects to be obtained.

Because it is same as the ACPD technique, the induced current also flows preferentially across the surface layer of metals (or metal specimens) based on the skin effect. The skin depth at which the current reduces to 1/*l* times lower than the current density at the surface can be expressed as shown in Equation (1). Additionally, the relationship between the potential drop and other properties can be expressed as shown in Equation (2).
(1)$$\delta =\frac{1}{\sqrt{\pi \cdot \mu \cdot \sigma \cdot f}}$$
where *μ* is the permeability (H/m), *σ* is the conductivity (1/Ωm), and *f* is the frequency (Hz). The potential drop Δ*V*, measured as shown in Figure 1, is a function of certain variables such as permeability, conductivity, current density, etc.
(2)$$\Delta V=I^{\prime }\rho \frac{1}{\delta \cdot b}=I^{\prime }\rho \frac{1}{S}=j\cdot \rho \cdot l$$
where Δ*V* is the potential drop (*V*), *I*’ is the induced current (A), *ρ* is the resistivity (Ωm), *j* is the current density (A/m^2^), *l* is the distance between the two measurement points of the induced current path (m), and *S* is the area of current flowing (m^2^).

### 2.2. Defects of Railway Wheel

The fatigue crack growth of railway wheels causes partial damage to the wheels or radial crack propagation resulting from rolling contact fatigue and braking friction heat. As a result, rail vehicle parts can be damaged or derailed. There are three main types of damage to wheels. A flat is damage that occurs when a wheel slides off a rail with braking force exceeding its limit. Owing to overheating of the wheel contacts and the subsequent rapid tempering process, the pearlitic steel at the contacts transforms into very brittle martensite.

Even small planes can cause fatigue defects or deform the wheel into an elliptical shape. Depending on the sliding situation, the tread is partially flattened in a circular or oval shape, or flattened in the shape of a connecting point, and plastic flow occurs continuously in a wavy shape or a band. As a rule, flats are accompanied by spalling. Spalling is a surface defect that occurs on part of the tread or on the entire circumference of a wheel due to rolling contact fatigue. It is also a phenomenon in which the thin surface layer moves in the direction of the load by repeatedly accumulating high stresses and traction forces on the tread of the wheel. That is, the plastic-flowing surface layer is strain-hardened and eventually damaged, resulting in cracks on the surface. When a crack occurs, spalling occurs on the surface caused by a series of loads and joins with the surrounding cracks. This spalling is easily triggered by overloaded wheel weight, insufficient strength of the wheel material, skids, and so on, but in most cases, it is caused by damage such as flats and thermal cracks. Thermal cracks are caused by friction in the brake shoes on the tread or flange. Thermal cracking is a state in which a lot of fine cracks are generated in the form of a net and develop in a relatively long manner in the axial direction. In particular, in the case of tread brake wheels, thermal cracks occur mainly due to the thermal effect of the outside of the brake shoe. Figure 2 shows the typical types of damage that occur on wheel surfaces. Figure 2a shows a defect that occurred below the surface of the wheel tread, and Figure 2b shows a thermal crack in the wheel.

## 3. Experimental Procedures

### 3.1. Measurement Apparatus

To detect the surface and subsurface defects of wheels, as shown in Figure 3, a defect detection system was installed using the induced AC potential drop technique. The system was composed of an automatic transfer jig through which specimens could be freely moved, a sensor for defect detection with a timer to maintain constant contact pressure with specimens, and a device to measure the variation in the potential drop. To measure changes in the potential drop, CGM-5R (Matelect Co., Harefield, UK) was used. The contact pressure of the sensor jig could be controlled manually or automatically, and it could be designed according to the sensor shape.

### 3.2. Test Conditions and Specimens

Table 2 summarizes the test conditions and specifications for the sensor and measurement unit. The electric current supplied was maintained at 2 A, and to obtain the optimum defect information, defect detection was conducted by changing the frequency and gain. The sensor used to detect defects was set up with an interval of 5 mm between the probes. Surface defects were made on the specimen by means of electric discharge machining, taking the form of semi-elliptical cracks. As shown in Figure 4, the surface defects were artificially created to test the system’s ability to detect defects with a depth of 0.5–2.0 mm. For internal defects, artificial defects were introduced this time with depths of ϕ1.5~2.0 mm according to the defect distance (dx) from the surface. Table 3 shows the mechanical properties of the wheel material. The wheel specimens were cut from the rim of a wheel, as shown in Figure 2.

Then, defects with widths (b) of 0.3 mm and 0.5 mm were produced in order to imitate fatigue defects as closely as possible. Table 4 demonstrates the positions and sizes of the artificial defects introduced to the specimen. In Table 1, a is the depth of the defect, c is the length of the defect, b is the width of the defect, dx is the distance between the surface of the specimen and the defect, and D1 and D2 are the diameters of the subsurface defects. When measuring a defect, the probe pin was placed at a right angle with the defect, and the variation in the potential drop was measured at intervals of 2.5 mm in the circumferential direction from the defect.

## 4. Experimental Results and Discussion

### 4.1. Surface Defects

An experiment was performed to test the surface and internal defects of the specimens, considering the defect width and the probe interval as influencing factors. Figure 5 and Figure 6 show the changes in the potential drop indicating surface defects. In the case of a probe interval of 5 mm, Figure 5 shows the measured results when the defect width was 0.5 mm at a gain of 70 dB, and Figure 6 demonstrates the measured results when the defect width was 0.3 mm at a gain of 90 dB. Figure 5 and Figure 6 show the common tendency for near-constant potential drops to be measured in positions with no defect, and in positions with a defect, for the potential drops to change greatly compared to those in positions with no defect. Furthermore, as the measurement probe moved closer to the end of the defect, the potential drops at the end of the defect gradually decreased, or if the position of the probe was on the defect, the potential drop increased at the position of the defect. This phenomenon is illustrated in detail in Figure 7. The results of the defect detection suggest that, in the case of a probe interval of 5 mm, it is possible to detect surface defects of the specimens until a defect depth of 0.5 mm is reached by using the induced AC potential drop technique. This means that, in terms of a surface defect depth of 0.5 mm or more, defects can be detected accurately, and it also means that, because the inspection cycle can be extended according to the degree of defect detection, the cost of inspection can be reduced. Moreover, the graph shows that the changes in the potential drop vary in line with defect depths from 0.5 mm to 2.0 mm, which means that the depth of the surface defect can be fully quantified by the induced AC potential drop technique. That is, an unknown surface defect size can be measured by using the induced AC potential drop method. Regarding the changes in the potential drop caused by the defect width, in Figure 5 and Figure 6, more notable changes are shown in the case of a defect width of 0.5 mm than a defect width of 0.3 mm. We propose that this should be considered when performing a detection experiment imitating a real defect. In the case of a defect width of 0.5 mm, since the changes in the potential drop are large, more accurate defect information can be obtained. Meanwhile, in the case of a defect width of 0.3 mm, the smaller the defect depth is, the narrower the range of changes in the potential drop.

Kwon et al. proved that, since it is easy for a narrow probe to detect a defect, it is possible to more intensively detect the current induced along the defect surface with a narrower probe than with a wider probe [23]. Therefore, it can be seen that, when using the induced AC potential drop technique, the narrower the interval of a probe is, the more accurately the defect can be measured. Figure 7 shows the potential drop distribution based on the position of the probe. Since a large difference is measured when the measuring terminal is located at the defect end, which is across the defect, and when the defect is located in the center of the measuring terminal, a small difference is measured, the potential drop distribution forms a ∧ shape. In this way, Figure 7 shows that the flow of the induced current changes in line with the edge effect at the defect end. In addition, it is known that the deeper the defect is, the bigger the potential drop at the defect end, meaning this effect relies on the depth of the defect. From this, it can be surmised that the potential drop around a defect changes as an indicator of the existence of the defect, and varies depending on the depth of the defect.

### 4.2. SubSurface Defects

As shown in Figure 8, 1.5 mm subsurface defects could be detected at positions 0.3~1.0 mm away from the surface, but the detection accuracy was reduced at positions more than 1.0 mm away from a defect. Figure 8 shows that 1.5 mm subsurface defects can be detected at positions 1.0 mm away from the surface using the induced AC potential drop technique. Figure 8 and Figure 9 show the distribution of changes in the potential drop for cracks of 2.0 mm and 1.5 mm. There is a height difference in the potential drop caused by the edge effect for larger defects, as shown in Figure 8. Yet, there is no height difference in the potential drop for smaller defects as the edge effect is offset. As shown in Figure 8 and Figure 9, it was possible to detect subsurface defects of 2.0 mm located 1 mm from the wheel surface using the concentrated induced potential difference method.

## 5. Conclusions

A defect inspection system using the induced AC potential drop technique was established to evaluate the possibility of detecting surface and subsurface defects on railway wheels. The results are as follows:(1)The present study tested the induced AC potential drop technique, determining that it was possible to detect surface defects up to 0.5 mm in depth and of 0.5 mm or larger on the surface of the wheel specimen.(2)It was possible to detect subsurface defects of 1.5 mm or more at a distance of 1.0 mm from the wheel surface.(3)Both surface and subsurface defects on the wheel material could be sensitively detected using the induced AC potential drop technique.

## Figures and Tables

**Figure 1 sensors-22-09981-f001:**
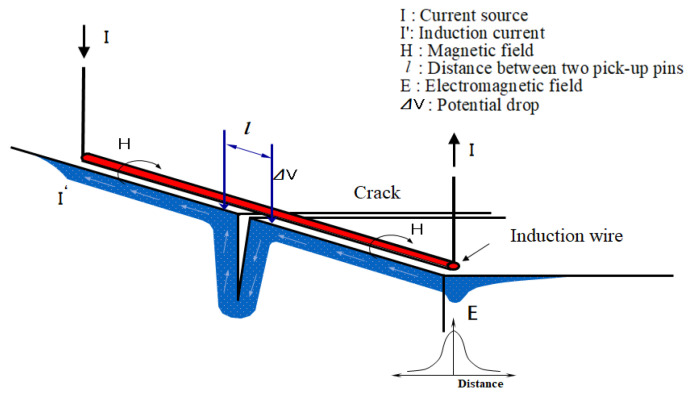
Principal of induced alternating current potential drop (IACPD).

**Figure 2 sensors-22-09981-f002:**
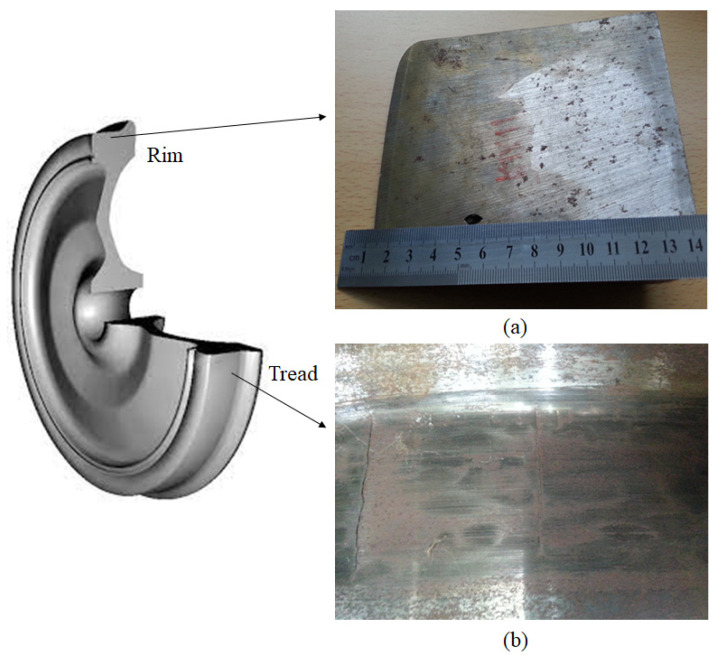
Defects in railway wheels: (**a**) subsurface crack; (**b**) surface crack.

**Figure 3 sensors-22-09981-f003:**
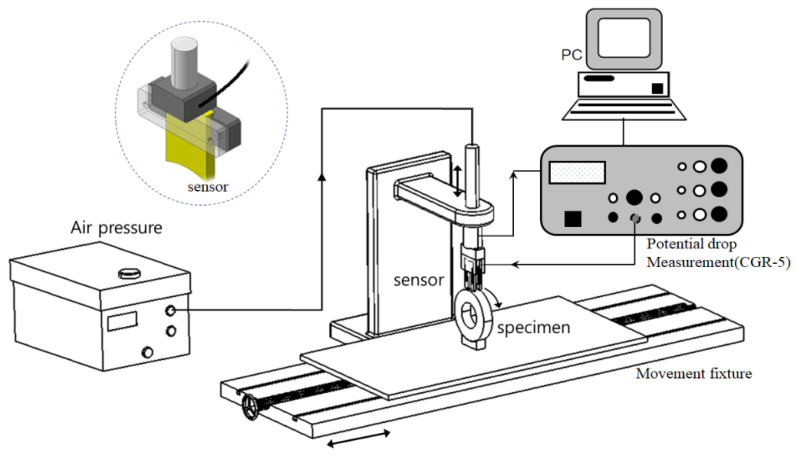
Measurement system.

**Figure 4 sensors-22-09981-f004:**
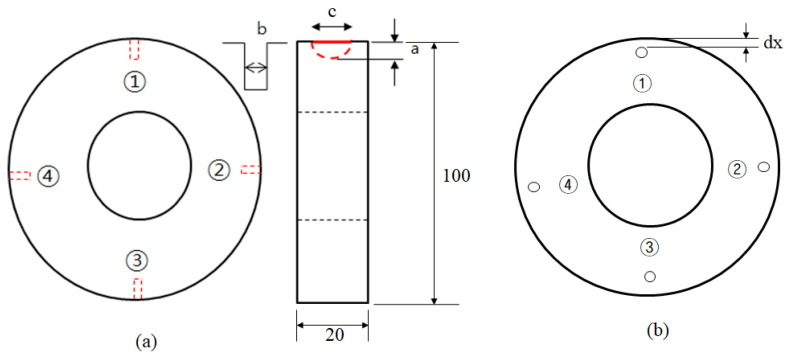
Geometry of specimens: (**a**) surface defect; (**b**) subsurface defect.

**Figure 5 sensors-22-09981-f005:**
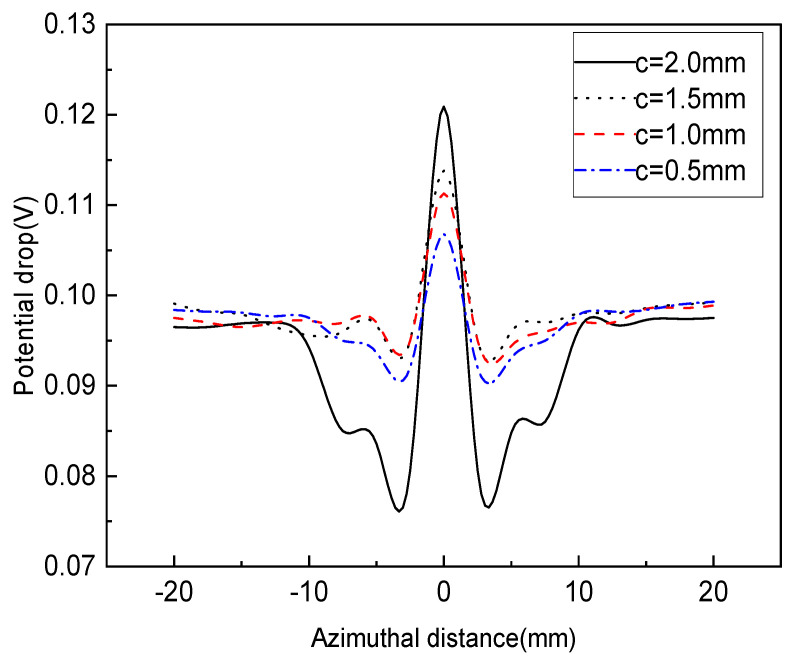
Variations of potential drop with different defect depths for a surface crack width of 0.5 mm.

**Figure 6 sensors-22-09981-f006:**
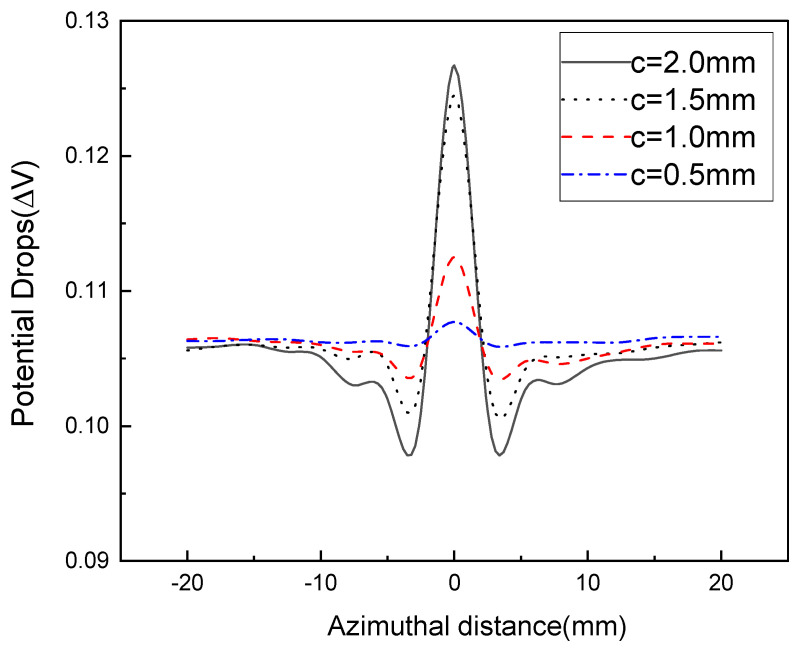
Variations of potential drop with different defect depths for a surface crack width of 0.3 mm.

**Figure 7 sensors-22-09981-f007:**
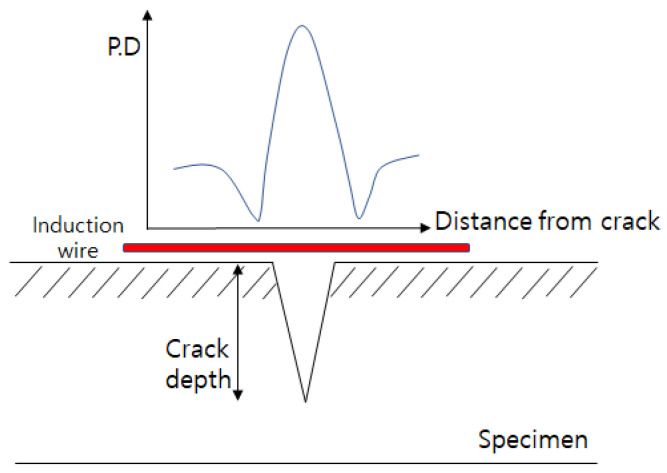
Potential drop distribution for crack.

**Figure 8 sensors-22-09981-f008:**
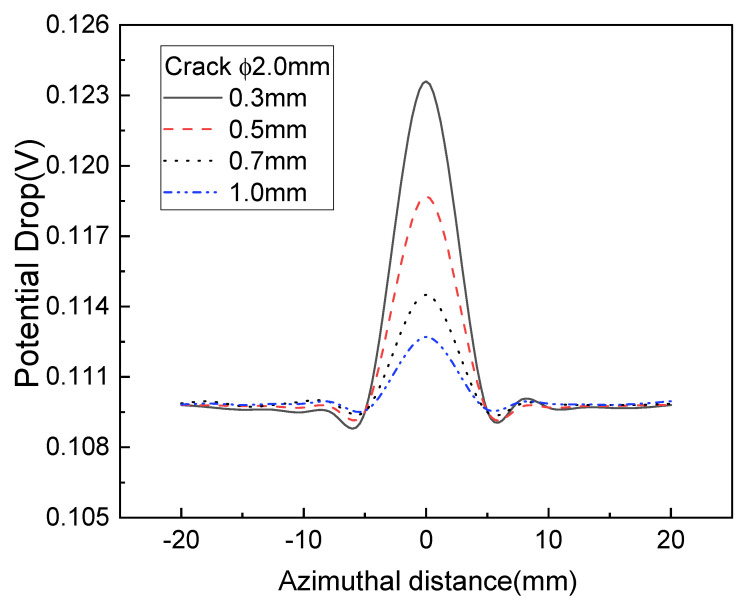
Variations of potential drop with different defect depths for a subsurface crack of 2.0 mm.

**Figure 9 sensors-22-09981-f009:**
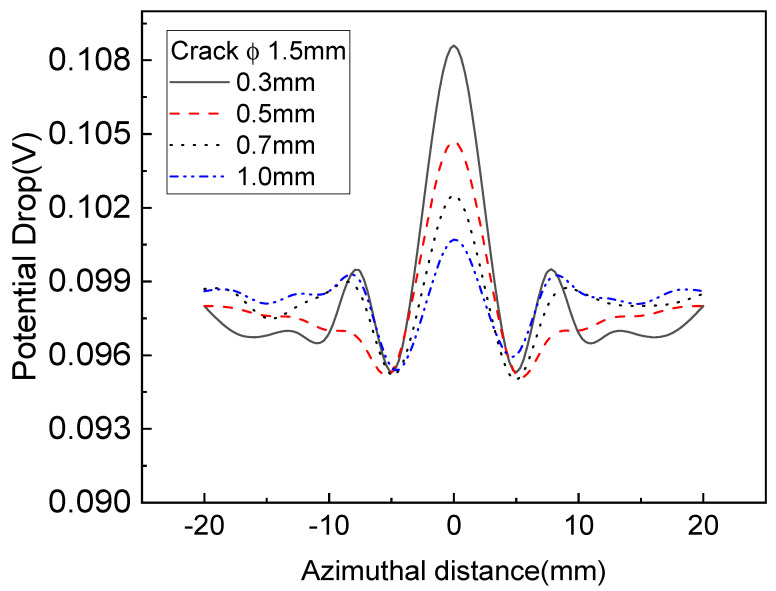
Variations of potential drop with different defect depths for a subsurface crack of 1.5 mm.

**Table 1 sensors-22-09981-t001:** Comparison of potential drop techniques.

	Method	A.C.		
Contents		IACPD ^(1)^	ACPD ^(2)^	ACFM ^(3)^
Current supply		Induction	Direction	Induction
No. of induction wires		1	-	4 or more
Probe pins		2	2	2
Distribution of PD		Concentration in measuring area	In surface	
Advantages		- Detection of surface and inner flaws - Concentration in measuring area - Easy measuring method - Auto measuring method	- Monitoring and measuring of surface - Small measurement system - No error due to heat by low current - Theoretical analysis is possible	
Disadvantages		-	- Error of predictions is large for surface cracks - Instability of surface current	

^(1)^ IACPD (induced alternating current potential drop). ^(2)^ ACPD (alternating current potential drop). ^(3)^ ACFM (alternating current field measurement).

**Table 2 sensors-22-09981-t002:** Test conditions and specifications of sensor and PD measurement unit.

Current (A)	Frequency (kHz)	Gain
2.0	3	70
Sensor	Distance of pick-up pins (mm)	5
	Length of induction wire (mm)	40
Potential drop measurement (CGR-5)	AC frequency (kHz)	0.3~100
	Gain (dB)	50~90
	AC current (A)	0.1~2

**Table 3 sensors-22-09981-t003:** Mechanical properties of wheel specimens.

Yield Strength (MPa)	Tensile Strength (MPa)	Elongation (%)	Hardness (Hv)
507	860	32	280

**Table 4 sensors-22-09981-t004:** Artificial cracks of wheel specimens.

	Crack	Surface (mm)		Subsurface (mm)		
Position		a	c	D1	D2	dx
①		2.0	8.0	2.0	1.5	0.3
②		1.5	6.0	2.0	1.5	0.5
③		1.0	4.0	2.0	1.5	0.7
④		0.5	2.0	2.0	1.5	1.0

## Data Availability

Not applicable.

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
