# Peer review of "Applicability Evaluation of Surface and Sub-Surface Defects for Railway Wheel Material Using Induced Alternating Current Potential Drops"

_sensors, 2022, doi:10.3390/s22249981_

Round 1

Reviewer 1 Report

Comments to the Author(s)

This manuscript presents an Applicability Evaluation on Surface and Sub-surface Defects for Railway wheel material using Induced Alternating Current Potential Drops. This paper contains a good effort related to Non-destructive evaluation systems based on Induced Alternating Current Potential Drops because they are widely used for crack depth measurements in metallic materials. This paper contains good material of interest to the NDT community. In general, the manuscript needs to be organized well. Some texts and figures in the manuscript need to be improved. The authors are suggested improving the paper by resolving the following issues (All the answers should be included in the manuscript):

1.      The authors shall clearly explain how this paper is an original contribution. What is the difference between the current work and the following works?

·         Alemi, A., Corman, F., & Lodewijks, G. (2017). Condition monitoring approaches for the detection of railway wheel defects. Proceedings of the Institution of Mechanical Engineers, Part F: Journal of Rail and Rapid Transit, 231(8), 961-981.

·         Shen, J. (2017). Responses of alternating current field measurement (ACFM) to rolling contact fatigue (RCF) cracks in railway rails (Doctoral dissertation, University of Warwick).

2.      The manuscript needs further improvement in the writing. It is recommended to be edited by an English editor.

3.      Page 1, line 17, instead of writing AC, please write it as “alternating current (AC)”

4.      Page 2, line 50, instead of writing AC (Alternating Current) electromagnetic field, please rewrite it as “alternating current (AC) electromagnetic field”

5.      Figure 1, please rewrite the caption of Figure 1 as “Principal of Induced Alternating Current Potential Drop (IACPD).

6.      Figure 1, I do not see the label (E ) in the schematic

7.      Page 3, lines 88, and 92, please remove the comma after the word “where”

8.      Page 6, line 212, please rewrite the following sentence” Potential drop ΔV, measured in the way shown in Fig. 1 is a function of many variables.” to “ The potential drop ΔV, measured in the way shown in Fig. 1, is a function of some variables.”

9.      Page 3, line 113, please add a comma after the word “cases” in this sentence “but in most cases it is usually...”

10.   Page 4, line 124, please correct the spelling of the word “measurement” in subtitle 3.1

11.  Page 3, line 113, I think the explanation of Fig. 2(a), and (b) are not matching the labels of Fig 2 and its caption.

12.   Fig 2, please rewrite the caption of Fig.2 as the following caption “Defects in the railway wheel: (a) sub-surface crack; (b) surface crack.”

13.   Figure 3, please rewrite the caption of Fig.3 with more details. Improve the Figure contains. For example, label the value and direction of AC.

14.  Page 5, line 138, please separate the value of current from its unit (2A)

15.  Page 5, line 153, the text “In Table 1, c is the depth of the defect, a is the 153 length of the defect, b is the width of the defect,”. However, in Fig.4, I see the (c) and (a) labels are reversed. Please explain that.

16.  Please separate the length values from its unit. For example, 5 mm.

17.   Fig. 5, Pleas rewrite the caption of Fig.5 to be “Variations of the potential drop with different defect depths for surface crack width of 0.5 mm.”. Also, rewrite the caption of Fig.6.

18.  Please explain the x-axis of Fig.5 and Fig.6. I see four different peaks in Fig. 6, please identify them in detail.

19.    Fig.5 and Fig. 6 represent the result for 0.5 mm and 0.3 mm defect widths, respectively. However, I see significantly different results. Please explain the differences between these two Figures.

20.   Please redraw Fig.8, and Fig.9 with clear color

21.   The conclusion needs to be improved.

Author Response

Reviewer 1;

  1. The authors shall clearly explain how this paper is an original contribution. What is the difference between the current work and the following works?

Alemi, A., Corman, F., & Lodewijks, G. (2017). Condition monitoring approaches for the detection of railway wheel defects. Proceedings of the Institution of Mechanical Engineers, Part F: Journal of Rail and Rapid Transit, 231(8), 961-981.

Shen, J. (2017). Responses of alternating current field measurement (ACFM) to rolling contact fatigue (RCF) cracks in railway rails (Doctoral dissertation, University of Warwick).

As point outed by reviewer, the manuscript is modified and Table 1 is added.

“Alemi et al reported extensively on the condition monitoring techniques of railway wheel in on-board and wayside measurement [21]. The authors stated that the necessarily of research on detection accuracy and monitoring for sub-surface defect. Shen et al discussed that the evaluation of surface fatigue crack using alternating current field measurement (ACFM) in rail [22].

Among many defect types occurring on the surface of railway wheels, although big defects are removed at the depot through wheel reprofiling, on regular inspection, the detection possibility of sub-surface defects in wheels decreases significantly.

The purpose of the present paper is to evaluate the possibility of detecting the surface and sub-surface defects of railway wheel material using the distribution of the induced alternating current (AC) electromagnetic field.”

  1. The manuscript needs further improvement in the writing. It is recommended to be edited by an English editor.

As suggested by reviewer, the present paper is edited by MDPI English service.

  1. Page1, line 17, instead of writing AC, please write it as “alternating current (AC)”

As suggested by reviewer, the sentences are rewritten.

“The results indicate that the technique can evaluate the possibility of detecting the surface and sub-surface defects of railway wheel specimens using the distribution of the alternating current (AC) electromagnetic field.”

  1. Page 2, line 50, instead of writing AC (Alternating Current) electromagnetic field, please rewrite it as “alternating current (AC) electromagnetic field”

As suggested by reviewer, the sentences are rewritten.

“The purpose of the present paper is to evaluate the possibility of detecting the surface and sub-surface defects of railway wheel material using the distribution of the induced alternating current (AC) electromagnetic field.”

  1. Figure 1, please rewrite the caption of Figure 1 as “Principal of Induced Alternating Current Potential Drop (IACPD)

As suggested by reviewer, the Fig.1 is rewritten.

Figure 1. Principal of Induced Alternating Current Potential Drop(IACPD)

  1. Figure 1, I do not see the label (E ) in the schematic

As suggested by reviewer, the Fig.1 is modified.

Figure 1. Principal of Induced Alternating Current Potential Drop(IACPD)

  1. Page 3, lines 88, and 92, please remove the comma after the word “where”

As suggested by reviewer, the sentences are modified as follow;

“where μ is the permeability (H/m), σ is the conductivity (1/ Ωm), and f is the frequency (Hz). The potential drop ΔV, measured as shown in Figure 1, is a function of some variables, such as permeability, conductivity, and current density, etc.”

“where ΔV is the potential drop (V), I΄ is the induced current (A), ρ is the resistivity (Ωm), j is the current density (A/m2), l is the distance between the two measurement points of the induced current path (m), and S is the area of current flowing (m2).”

  1. Page 3, line 212, please rewrite the following sentence” Potential drop ΔV, measured in the way shown in Fig. 1 is a function of many variables.” to “ The potential drop ΔV, measured in the way shown in Fig. 1, is a function of some variables.”

As suggested by reviewer, the sentences are modified as follow;

“The potential drop ΔV, measured as shown in Figure 1, is a function of some variables, such as permeability, conductivity, and current density, etc.”

  1. Page 3, line 113, please add a comma after the word “cases” in this sentence “but in most cases it is usually...”

As suggested by reviewer, the sentences are modified as follow;

“This spalling is easily caused by overloaded wheel weight, insufficient strength of the wheel material, skids, and so on, but in most cases, it is usually caused by damage such as flats and thermal cracks.”

  1. Page 4, line 124, please correct the spelling of the word “measurement” in subtitle 3.1

As suggested by reviewer, the word is modified

“3.1. Measurement apparatus”

  1. Page 3, line 113, I think the explanation of Fig. 2(a), and (b) are not matching the labels of Fig 2 and its caption.

As suggested by reviewer, the sentences are modified.

“Figure 2a shows a defect that occurred below the surface of the wheel tread and Figure 2b shows a thermal crack in the wheel.”

  1. Fig 2, please rewrite the caption of Fig.2 as the following caption “Defects in the railway wheel: (a) sub-surface crack; (b) surface crack.”

As suggested by reviewer, the Fig.2 is rewritten.

“Figure 2. Defects in railway wheel: (a) sub-surface crack; (b) surface crack.”

  1. Figure 3, please rewrite the caption of Fig.3 with more details. Improve the Figure contains. For example, label the value and direction of AC.

As suggested by reviewer, the Fig.3 is redrawn.

Figure 3. Measurement system.

  1. Page 5, line 138, please separate the value of current from its unit (2A)

As suggested by reviewer, the unit is rewritten.

“Table 1 demonstrates the summary of specimen conditions. The electric current supplied was maintained at 2 A, and to obtain the optimum defect information, the defect detection was conducted by changing frequency and gain.”

  1. Page 5, line 153, the text “In Table 1, c is the depth of the defect, a is the 153 length of the defect, b is the width of the defect,”. However, in Fig.4, I see the (c) and (a) labels are reversed. Please explain that.

As point outed by reviewer, the sentences and Table 2 are modified

“In Table 1, a is the depth of the defect, c is the length of the defect, b is the width of the defect, dx is the distance between the surface of the specimen and the defect, and D1 and D2 are the diameters of the sub-surface defects.”

  1. Please separate the length values from its unit. For example, 5 mm.

As suggested by reviewer, the sentences are modified

“The results of the defect detection suggest that in the case of a probe interval of 5 mm, it is possible to detect the surface defect of the specimen until a defect depth of 0.5 mm by using the induced AC potential drop technique.”

  1. Fig. 5, Pleas rewrite the caption of Fig.5 to be “Variations of the potential drop with different defect depths for surface crack width of 0.5 mm.”. Also, rewrite the caption of Fig.6.

As suggested by reviewer, the captions of Fig.5 and Fig.6 are modified.

“Figure 5. Variations of potential drop with different defect depths for a surface crack width of 0.5 mm.

Figure 6. Variations of potential drop with different defect depths for a surface crack width of 0.3mm.”

  1. Please explain the x-axis of Fig.5 and Fig.6. I see four different peaks in Fig. 6, please identify them in detail.

As point outed by reviewer, the Fig.6 is modified.

“In Fig.5 and Fig.6, the x-axis is the measurement distance in the azimuthal direction of specimen. “

  1. Fig.5 and Fig. 6 represent the result for 0.5 mm and 0.3 mm defect widths, respectively. However, I see significantly different results. Please explain the differences between these two Figures.

As point outed by reviewer, the Fig.6 is modified.

The defect width of 0.3 mm and 0.5 mmm is artificially made by Electric Discharge Machining. Authors would like to simulate the actual defect width. The defect width of 0.3 mm and 0.5 mm is not clear remarkably under test condition of 3kHz, 70dB. In Fig. 5, the P.D is measured at gain 90dB.

Figure 6. Variations of potential drop with different defect depths for a surface crack width of 0.3mm

  1. Please redraw Fig.8, and Fig.9 with clear color

As suggested by reviewer, the Fig.8 and Fig.9 is modified.

Figure 8. Variations of potential drop with different defect depths for a sub-surface crack of 2.0 mm.

Figure 9. Variations of potential drop with different defect depths for a sub-surface crack of 1.5mm.

  1. The conclusion needs to be improved.

As point outed by reviewer, the conclusion is modified.

“The defect inspection system using the induced AC potential drop technique was established to evaluate the possibility of detecting surface defects and sub-surface defects on railway wheels. The results are as follows.

1) The present study used the induced AC potential drop technique to find that it was possible to detect defects up to 0.5mm in depth, and 0.5mm surface defects or larger were definitely detected on the surface of the wheel specimen.

2) In the case of sub-surface defects, it was possible to detect sub-surface defects of 1.5 mm or more at a distance of 1.0 mm from the wheel surface.

3) For surface defects and sub-surface defects on the wheel material, the defects were able to be sensitively detected using the induced AC potential drop technique.”

Reviewer 2 Report

The paper discussed an interesting NDT technique for defect sizing. The comments are listed below:

1.      The advantages of induced ACPD over the conventional ACPD method were not well discussed. According to the setup information in section 3.1, the sensor still needs direct contact with the specimen surface. Is it able to work in a non-contact way?

2.      More details should be provided for the measurement apparatus, such as the sensor model, measurement system (detail parameters), measurement spacing, etc.

3.      The defect information is a little confusing (Table 2 and Figure 4); what is the width of the four defects? What are D1 and D2? How are the surface defects and subsurface defects generated?

4.      According to Figure 5, the measured defect width seems to be much larger than the real width. How to use the PD to determine the defect size (width)? Is there a correlation between PD and crack depth?

5.      Why does the 0.3 mm defect have the largest PD amplitude? 

Author Response

Reviewer 2;

  1. The advantages of induced ACPD over the conventional ACPD method were not well discussed. According to the setup information in section 3.1, the sensor still needs direct contact with the specimen surface. Is it able to work in a non-contact way?

As point outed by reviewer, the sentence is added.

“The conventional ACPD method is that the current flows through the specimen, IACPD method is that the current flows through the induction wire and induced current is detected.

When a current is applied to an induction wire, an electromagnetic field is induced in the area surrounding the induction wire. If alternating current flows in an induction wire placed near a conductive metal, the electromagnetic field induces a current in the conductive metal. Accordingly, the potential drop associated with the induced current can be measured.

  1. More details should be provided for the measurement apparatus, such as the sensor model, measurement system (detail parameters), measurement spacing, etc.

As suggested by reviewer, the Fig. 3 and Table 2 are modified.

Figure 3. Measurement system.

Table 2. Test conditions and specification of sensor and P.D measurement unit.

Current(A)

Frequency(kHz)

Gain

2.0

3

70

Senor

Distance of pick-up pins (mm)

5

Length of induction wire (mm)

40

Potential drop

Measurement (CGR-5)

AC frequency (kHz)

0.3~100

Gain (dB)

50~90

AC current (A)

0.1~2

  1. The defect information is a little confusing (Table 2 and Figure 4); what is the width of the four defects? What are D1 and D2? How are the surface defects and subsurface defects generated?

As suggested by reviewer, the sentence is modified.

“In Table 1, a is the depth of the defect, c is the length of the defect, b is the width of the defect, dx is the distance between the surface of the specimen and the defect, and D1 and D2 are the diameters of the sub-surface defects.”

“The surface defects were inserted into the specimen by means of electric discharge machining to perform a defect detection test, and the shape of the defect was processed into semi-elliptical cracks. As shown in Figure 4, the surface defects were processed using artificial defects to detect defects with a defect depth of 0.5 mm to 2.0 mm. For internal defects, artificial defects were processed to detect defects with depths of Ï•1.5 mm~2.0 mm according to the defect distance (dx) from the surface.”

  1. According to Figure 5, the measured defect width seems to be much larger than the real width. How to use the PD to determine the defect size (width)? Is there a correlation between PD and crack depth?

As suggested by reviewer, the Fig. 5 is modified.

The defect width and depth are artificially made by Electric Discharge Machining.

When measuring the defect, the probe pin was placed at a right angle for the defect of the wheel, and the variation in the potential drop was measured at intervals of 2.5 mm in the circumferential direction from the defect of the wheel.

Authors would like to investigate the correlation between PD and crack depth as a future work.

Figure 5. Variations of potential drop with different defect depths for a surface crack width of 0.5 mm.

  1. Why does the 0.3 mm defect have the largest PD amplitude?

As suggested by reviewer, the sentence is modified.

The defect width of 0.3 mm and 0.5 mmm is artificially made by Electric Discharge Machining. Authors would like to simulate the actual defect width. The defect width of 0.3 mm and 0.5 mm is not clear remarkably under test condition of 3kHz, 70dB. In Fig. 5, the P.D is measured at gain 90dB.

Figure 5 shows the measured results when the defect width was 0.5 mm at gain of 70dB, and Figure 6 demonstrates the measured results when the defect width was 0.3 mm at gain of 90dB.

“In Fig. 6, the P.D is measured at gain 90dB.”

Figure 6. Variations of potential drop with different defect depths for a surface crack width of 0.3mm

Round 2

Reviewer 1 Report

The authors have addressed the reviewer's concerns. And thus can be published on journal.

Reviewer 2 Report

The revision of the manuscript is sufficient to improve the quality of the paper, and it is ready to be published in its present form. The reviewer does not have further comments on the manuscript.